# Impact of the Stress Response on Quaternary Ammonium Compound Disinfectant Susceptibility in *Serratia* Species

**DOI:** 10.3390/microorganisms12112240

**Published:** 2024-11-06

**Authors:** Samantha McCarlie, Robert R. Bragg

**Affiliations:** Department of Microbiology and Biochemistry, University of the Free State, Bloemfontein 9301, South Africa; samanthamccarlie@gmail.com

**Keywords:** antimicrobial resistance, quaternary ammonium compounds, polyamines, RNA-Seq

## Abstract

The well-known problem of antibiotic resistance foreshadows a similar threat posed by microbial resistance to biocides such as disinfectants and antiseptics. These products are vital for infection control, yet their overuse during the COVID-19 pandemic has accelerated the development of resistant microorganisms. This study investigates the molecular mechanisms underlying disinfectant resistance in *Serratia* sp. HRI. The transcriptomic responses of *Serratia* sp. HRI were used to identify significant gene expression changes during exposure to QACs and revealed increased methionine transport and polyamine synthesis. Polyamines, crucial in cellular stress responses, were notably upregulated, suggesting a pivotal role of the stress response in disinfectant resistance. Further, our susceptibility tests revealed a marked decrease in susceptibility to QACs under various stress conditions, supporting the hypothesis that stress responses, mediated by polyamines, decrease susceptibility to QACs. This research highlights polyamines as key players in disinfectant resistance, offering novel insights into resistance mechanisms and antimicrobial susceptibility. Our findings emphasise the need for continued investigation into disinfectant resistance and the role of stress responses, particularly polyamine-mediated mechanisms, to direct strategies for preserving disinfectant efficacy and developing future antimicrobial agents.

## 1. Introduction

Disinfectants are crucial for infection control and preventing food-borne illnesses in healthcare, agriculture, and the food industries. The COVID-19 pandemic highlighted our reliance on biocides, such as disinfectants and antiseptics, as part of biosecurity and infection control. During the pandemic, the use of disinfectants increased drastically, creating greater selective pressure for resistance development [1,2]. Additionally, the use of poor-quality disinfectants during the pandemic has led to a rise in the rates of disinfectant-resistant microorganisms [3,4]. Currently, the most significant impact of resistance to disinfectants can be seen in the rapidly rising rates of hospital-acquired infections around the globe due to impaired infection control procedures and products [3]. 

*Serratia marcescens* has emerged as an opportunistic nosocomial pathogen causing various types of infections [5] and has gained a notorious reputation for causing hospital-acquired infections in paediatric and neonatal wards [6,7,8,9,10]. This opportunistic pathogen has been isolated on several occasions from bottles of disinfectants used in hospitals, revealing its ability to survive within bottles of disinfectants [9,11,12,13]. Its ability to survive and cause infection within the healthcare setting has been attributed to intrinsic and acquired antimicrobial resistance within this genus [5,14,15,16]. 

Current research into antimicrobial resistance has focused on antibiotics, with resistance to disinfectants as a relatively novel field of study that has gained more attention post-COVID-19. In light of this, our understanding of the molecular mechanisms of disinfectant resistance is still poor, and current hypotheses require more evidence. Once the mechanisms of the resistance phenotype are understood, this knowledge can be used to safeguard current disinfectants and direct the design of future antimicrobial compounds.

Quaternary ammonium compounds (QACs) were the largest group of disinfectants approved for use during the COVID-19 pandemic by the EPA [17]. They function by targeting the cell membrane of microorganisms and result in the formation of reactive oxygen species (ROS), which cause further cellular damage [18,19]. QACs remain the focus of most studies on resistance to disinfectants, and slowly, the molecular mechanisms are being uncovered [20,21,22,23]. Gene expression studies are gaining traction in the study of antimicrobial resistance as RNA-Seq techniques advance, becoming more accurate and reliable, and further validation is only required at very low levels of differential expression [24]. This robust technique reveals which systems and individual genes are most differentially expressed upon exposure to antimicrobials and uncovers insights into resistance mechanisms.

In industries dependent on infection and quality control, physical agents are used to hinder microbial growth, such as high or low temperatures, desiccation, and osmotic pressure [25,26]. Chemical agents include disinfectants and control microbial growth through widespread membrane and oxidative damage [18,19]. These adverse conditions activate the bacterial stress response, which can affect susceptibility to antibiotics [27,28,29,30,31,32]. Recent work highlighted the stress response as a key system differentially expressed after exposure to QAC disinfectants [33]. However, little to no work has been completed to evaluate the effect of stressors on disinfectant susceptibility and resistance.

In this study, disinfectant-resistant isolate *Serratia* sp. HRI [34] was used to elucidate the molecular mechanisms of QAC resistance through transcriptomics. Thereafter, the impact of the stress response on the susceptibility to a QAC product was evaluated together with the most closely related type strain, *Serratia marcescens* ATCC 13880. 

## 2. Materials and Methods

*Serratia* sp. HRI exhibits high QAC resistance and thus provides a unique opportunity to study bacterial resistance to QAC-based disinfectants [34]. As a control, the most closely related type strain, *Serratia marcescens* ATCC 13880, is used as a susceptible strain for comparison [34].

### 2.1. Microorganisms

The bacteria used in this study are highly resistant isolate *Serrata* sp. HRI [34] from the University of the Free State culture collection and the most closely related type strain, *Serratia marcescens* subsp. marcescens ATCC 13880, obtained from the American Type Culture Collection. *Serratia* sp. HRI was selected for study due to high levels of QAC resistance, providing an opportunity to elucidate QAC-resistance mechanisms [34].

### 2.2. Disinfectants

For this study, two disinfectants were used: Virukill™ (Didecyldimethylammonium Chloride 120 g/L) and DDAC (800 g/L Didecyldimethylammonium Chloride) were obtained from ICA International Chemicals, Stellenbosch, South Africa. The active ingredient DDAC was used to evaluate QAC susceptibility for only the active ingredient (pure QAC), whereas Virukill^®^ is a registered product, with additional additives, and therefore, represents a commonly used QAC-based infection control product.

### 2.3. Bacterial Cultivation for RNA-Seq

A pre-inoculum of *Serratia* sp. HRI was prepared by transferring a single colony from 24 h agar plates to 25 mL Luria Broth (LB) medium (10 g/L Merck Tryptone, 10 g/L Merck sodium chloride, 5 g/L Merck yeast extract, Darmstadt, Germany) in 250 mL Erlenmeyer flasks and incubated at 37 °C for 16 h. From this pre-inoculum, 2.5 mL was added into 500 mL Erlenmeyer flasks containing 47.5 mL LB media and incubated at 250 r min^−1^ and 37 °C for 60 min to ensure cultures would enter an exponential phase of growth based on previous studies [35]. After 60 min, Minimal Inhibitory Concentration (MIC) levels of DDAC (76.62 mg/L, South Africa) or Virukill™ (1200 mg/L, South Africa), determined previously [34], were added to the test cultures, and a contact time of 20 min was allowed before the cell pellets were immediately treated with Invitrogen RNA*later* Stabilization Solution (Carlsbad, CA, USA) to store and stabilise RNA within the cells before RNA extraction commenced.

### 2.4. RNA Extraction and Quality Control

RNA extraction was completed on fresh cell pellets using the RNeasy mini kit (Qiagen, Hilden, Germany) according to the manufacturer’s instructions in biological triplicates for untreated (control) and disinfectant-treated samples. An additional on-column DNAse digestion was performed to eliminate possible genomic DNA (gDNA) contamination using the ZymoResearch (Irvine, CA, USA) DNase I set per the manufacturer’s instructions. RNA quantity and quality were evaluated on agarose gels, a NanoDrop 1000 spectrophotometer and the Qubit RNA BR Assay Kit (Thermo Fisher Scientific, Waltham, MA, USA). Once all samples had passed QC based on RNA integrity, concentration, and purity, the samples were sent to the Centre for Proteomic and Genomic Research for sequencing (Cape Town, South Africa) and were prepared as described below.

Before ribosomal depletion, the total RNA for all samples was precipitated by adding sodium acetate (0.3 M) and 3 volumes of ice-cold absolute ethanol. The RNA was allowed to precipitate at −80 °C overnight. RNA was pelleted by centrifugation at 13,000× *g* for 10 min. RNA pellets were washed twice with ice-cold 70% ethanol and centrifuged at 13,000× *g* for 10 min. Ethanol was removed, and the pellets were dried at room temperature. The pellets were then resuspended in nuclease-free water. RNA was re-evaluated for purity using the NanoDrop 8000 spectrophotometer, quantity using the Qubit RNA BR Assay Kit (Waltham, MA, USA), and integrity using the Agilent Bioanalyzer RNA 6000 Nano Kit (Waldbronn, Germany). 

Total RNA extracts underwent Ribosomal depletion, RNA-Seq library preparation and QC, RNA-Seq library pooling, and RNA-Seq library sequencing as elaborated in Appendix A.

### 2.5. RNA-Seq Data Analysis

RNA-seq data were obtained from bacterial species following sequencing on Illumina (San Diego, CA, USA). The raw sequencing reads were quality-checked using FastQC. The adapter sequences were removed using Trimmomatic as part of the Trinity pipeline. Trinity (version 2.15.1) software was used for de novo transcriptome assembly with the default parameters. The BUSCO software (available at https://busco.ezlab.org/) was executed to assess the completeness of the de novo assembly using a reference bacterial dataset from OrthoDB (bacteria_odb10). Upon completion of the BUSCO analysis, the output directory contains reports and metrics detailing the completeness of the transcriptome assembly. Key metrics such as the percentage of complete single-copy and duplicated BUSCOs, fragmented BUSCOs, and missing BUSCOs were evaluated to assess the quality and completeness of the assembly. The UniProt database (2024_02 release) of reviewed proteins (Swiss-Prot) was used as the reference database for alignment using diamond (v2.1.9) software. Transdecoder (v5.7.1) was used to predict ORFs and the protein-coding sections of the identified transcripts in the Trinity transcriptome. The predicted protein-coding frames from TransDecoder were used as input to align to the UniProt database.

The newly assembled transcriptome, in FASTA format, was indexed using Bowtie2-build to create a Bowtie2 index. The indexed transcriptome was used as the reference for aligning RNA-seq reads using Bowtie2. Paired-end reads from the RNA-seq data were aligned to the transcriptome. The RSEM software (version 1.3.1) was employed to estimate transcript abundances based on the aligned RNA-seq reads. The edegR package was used to determine the *p*-value and False Discovery Rate (FDR) for the given transcripts for all the different conditions and positive control samples. Lowly expressed transcripts were removed, and the TMM normalisation technique was employed as part of the edgeR package. Since the samples were technical replicates, a constant dispersion estimate of 0.01 was used during the DGE analysis. A volcano plot was used for visualisation. The graphs were generated by using R and custom scripts. This approach applied thresholds to DEG, including log fold change (logFC) ≥ 1 or ≤−1 (corresponding to 2-fold upregulation or downregulation, respectively), *p*-value < 0.05, and FDR < 0.05.

### 2.6. Minimal Inhibitory Concentration Susceptibility Tests

Minimal inhibitory concentrations (MICs) for Virukill™ (South Africa) were determined before and after stress treatments for *Serratia* sp. HRI and *Serratia marcescens* ATCC 13,880. MICs were performed by the broth dilution method [36], which started with a fresh overnight culture that had been standardised to the McFarland standard of 0.5. The disinfectants were serially diluted 2-fold with sterile water in a 96-well microtitre plate, and a 10% inoculum of the bacterium was added to the disinfectant dilutions. The 96-well plates were incubated at 37 °C for a 20 min contact time. Thereafter, a 10% inoculum was transferred into a new 96-well microtitre plate containing BHI growth media and incubated at 37 °C overnight. The MIC was determined to be the lowest concentration at which no visible bacterial growth occurred. The average MIC values were calculated from at least 4 technical repeats and a minimum of 3 biological replicates for each bacterium against each disinfectant.

### 2.7. Selected Stress Factor Procedures

All bacterial cultures were prepared overnight in a pre-inoculum in BHI growth media before being subjected to stress treatments in biological triplicates. The bacterial cultures were subject to one of seven different stress treatments and standardised to the McFarland standard before the post-treatment MICs were completed. Osmotic stress was induced by the addition of 3.5% NaCl to growth media for 24 h. Cold stress was induced by storing cultures at 4 °C for 48 h. Acidic conditions were created by adjusting the pH of the growth media to pH 5.5 for 30 h. Alkaline conditions were brought about by adjusting the pH of the growth media to pH 8.5 for 30 h. Heat stress was induced by incubating cultures at 55 °C for 20 min. Envelope stress was induced by the addition of 9.8 mg/mL benzalkonium chloride and 0.098 mg/mL benzalkonium chloride to *Serratia* sp. HRI and *Serratia marcescens* ATCC, respectively, for 40 min. Lastly, oxidative stress was induced by adding 0.1 mm H_2_O_2_ for 20 min. 

## 3. Results

### 3.1. RNA-Seq Differential Expression Analysis

#### 3.1.1. *Serratia* sp. HRI Treated with DDAC Disinfectant Compared to Untreated Controls

Figure 1 and Table 1 depict the DGE patterns and results between the table.

AC-treated bacteria and untreated bacterial samples. Differential gene expression patterns in Figure 1 depict a large number of genes differentially expressed after treatment with the majority of differential expression being upregulation. The most significantly differentially expressed genes were labelled in Figure 1, and details are presented in Table 1. The gene functions annotated in Table 1 revealed that treatment with the DDAC disinfectant led to an increase in the expression of ABC transporters associated with methionine import, along with an increased expression of genes associated with protein and polyamine synthesis, as well as increased options for substrate use as a carbon source.

#### 3.1.2. *Serratia* sp. HRI Treated with Virukill™ Disinfectant Compared to Untreated Controls

Figure 2 and Table 2 depict the DGE patterns between Virukill™-treated bacteria and untreated bacterial samples. Differential gene expression patterns in Figure 2 depict a large number of genes differentially expressed after treatment with the majority of differential expression being upregulation. The most significantly differentially expressed genes were labelled in Figure 2, and details are presented in Table 2. The gene functions annotated in Table 2 revealed that Virukill^®^-treated bacteria upregulated genes associated with amino acid and protein synthesis and ABC transporters linked to methionine import.

#### 3.1.3. *Serratia* sp. HRI DDAC Treatment Compared to Virukill™ Treatment

Figure 3 and Table 3 depict the DGE patterns and genes with the most significant differential expression between the DDAC- and Virukill™-treated samples. These genes are most different in the bacterial response to the two disinfectant treatments. Increased expression was seen in genes associated with polyamine synthesis, general protein and carbohydrate transport, and putrescine and ascorbate transport. In addition, an increase in substrate utilisation for a carbon source was seen.

Although Virukill^®^ utilises DDAC as its active compound, as a final product used in industry, it contains many additional compounds in its formulation. Therefore, any changes in susceptibility and DGE profiles can be attributed to the modified formulation and additives in the final disinfectant product.

RNA-Seq data in Table 1, Table 2 and Table 3 revealed strong differential expression in polyamine synthesis and transport. An analysis of the literature elucidated the links between polyamines and the stress response in reducing susceptibility to antibiotics. However, a lack of research has been completed to reveal whether this is the case for disinfectant antimicrobials as well. Therefore, susceptibility tests were completed to determine what effect the stress response has on disinfectant susceptibility.

### 3.2. Effect of Stressors on Disinfectant Susceptibility

*Serratia* sp. HRI (resistant) and *Serratia marcescens* ATCC (type strain) were exposed to seven different stress conditions before and after susceptibility tests were completed to determine if any changes in susceptibility occurred after exposure to a stressor.

Susceptibility tests for the resistant isolate are shown in Figure 4, and *Serratia* sp. HRI showed significant increases (99% confidence) in MICs for the QAC-based disinfectant after all seven stressors.

When assessing the type strain, MICs for the QAC-based product in Figure 5 increased after cold stress, and the remaining six stressors increased the MIC significantly (99% confidence). For both the resistant isolate and type strain, MICs against QAC-based disinfectant Virukill™ increased significantly after exposure to stressors. 

## 4. Discussion

Analysis of gene expression by RNA-Seq is a powerful tool to uncover novel molecular resistance mechanisms [30]. However, as RNA-Seq focuses on quantifying transcripts, gene expression results may not always translate to a phenotypic response [37,38]. In this work, we have identified possible molecular mechanisms of resistance at a gene expression level and used susceptibility tests to test this hypothesis on a phenotypic level. 

The transcriptomic analysis results summarised in Table 1, Table 2 and Table 3 revealed the genes with the most significant differential expression during treatment with two QAC-based disinfectants, namely, Virukill^®^ and DDAC. Transcripts from the DDAC- and Virukill^®^-treated samples were characterised by increased methionine transport as part of the ABC transporter complex and increased polyamine synthesis. DDAC-treated samples were further differentiated from Virukill™-treated samples by an additional five-fold increase in polyamine synthesis. Differences in DGE between DDAC and the product Virukill^®^ can be attributed to additives and modifications in the final product. These differences emphasise the importance of research focusing on final products in use instead of isolated active ingredients. A similar upregulation of polyamine transport was observed in *Streptococcus pneumoniae* in response to hydrogen peroxide treatment and physiological temperature stress [39].

During treatment with QAC disinfectants, ROS are produced [18,19], which leads to the oxidation of methionine (Met) residues in proteins, resulting in altered protein structures and impaired protein function [40]. Therefore, the upregulation of methionine transporters could replenish methionine oxidised by ROS to maintain proteostasis. Additionally, there is an integral connection between methionine and polyamine metabolism through S-adenosylmethionine (SAM) [41,42]. Methionine is the precursor for SAM, a crucial substrate for the synthesis of polyamines [43,44]. Polyamines, such as spermidine, spermine, and putrescine, are essential for various cellular functions, including cell growth, division, and the stress response [45]. Polyamine synthesis (upregulated in the transcriptomic data) involves a series of enzymatic reactions starting from SAM [43,44,45]. Methionine availability, which is impacted by methionine import (upregulated here), influences polyamine synthesis, and polyamine levels can impact methionine metabolism through feedback mechanisms [43,44].

Polyamines are simple primordial stress molecules [43]. They bring about reduced susceptibility to antibiotics by affecting permeability across the cell membrane, thereby modulating antibiotic uptake, binding to antibiotics interfering with the action of fluoroquinolones, influencing bacterial virulence by affecting biofilm formation and toxin production, acting as free radical scavengers, and lastly, influencing genes involved in antibiotic resistance [32,44,45,46,47]. However, this effect does differ between bacterial species, the type of antibiotic, and the concentration of polyamines [32]. Polyamines have been linked to susceptibility and resistance to antibiotics. However, little information exists on the effect of polyamines on susceptibility and resistance to disinfectants and other biocides. A recent paper by Short and co-workers [48] described an *Acinetobacter baumannii* disinfectant resistance protein, AmvA, as a spermidine and spermine polyamine efflux pump for the first time. This supports the notion that polyamines, through the stress response, affect susceptibility to disinfectant antimicrobials. However, further research is needed to confirm the link between the stress response and disinfectant susceptibility.

Differences in susceptibility and DGE profiles for DDAC and Viurkill^®^ highlight the need to do research on antimicrobial susceptibility to final products. Research focusing on products that are currently in use in industry allows for the results to be applicable to industry and “real-world” environments. Many antimicrobials are used as part of formulations in final products, and as such, any research that hopes to be relevant to hospital environments or agriculture needs to be completed on the products currently used in these environments. Therefore, further susceptibility tests were completed using the final product and with stressors as would be applicable environmentally.

The stress MICs in Figure 4 and Figure 5 address this research gap, using *Serratia* sp. HRI, a resistant isolate and *Serratia marcescens* ATCC as the most closely related type strain. MIC tests were conducted for a QAC-based Virukill™ disinfectant. The susceptibility of the resistant isolate to the QAC disinfectant decreased significantly for all seven stressors, with each stress condition leading to increased resistance to the QAC disinfectant. In particular, osmotic and oxidative stress conditions resulted in a 60-fold increase in resistance to the disinfectant, as depicted in Figure 4. This trend was also observed in the type strain, with significant decreases in susceptibility to the QAC disinfectant for all stress conditions except cold stress, which decreased susceptibility but did not reach the significance threshold. Thus, these results show that stress conditions decrease susceptibility significantly to the QAC disinfectant for both bacteria and suggest that the stress response is responsible for up to 60-fold reduced susceptibility to the QAC.

The literature suggests that susceptibility linked to the stress response differs between disinfectant active ingredients. In a recent study on uropathogenic *Escherichia coli*, the oxidative stress response was induced after exposure to benzalkonium chloride (QAC), hydrogen peroxide, and triclosan but not chlorhexidine, resulting in different stress responses between biocides [49].

Together, the transcriptomic data and susceptibility tests point to the stress response as a major player in susceptibility and resistance to disinfectants, modulated by polyamines, as observed in the gene expression profiles of Table 1, Table 2 and Table 3. When considering the impact of polyamines on antimicrobial resistance, the role of polyamines in enhancing antioxidant defence mechanisms and subsequent oxidative stress tolerance in plants and mammals is evident [43]. However, there is a research gap in the impact of polyamines and the stress response on disinfectant susceptibility.

Proposed mechanisms on how polyamines link to reduced susceptibility and resistance to disinfectants and biocides are cell membrane protection, free radical scavenging, and modulation of the stress response. Many disinfectants (such as QACs) function by disrupting the cell membrane. With their positively charged amine groups, polyamines interact with negatively charged phospholipids and stabilise the cell membrane. Additionally, polyamines can interact with membrane proteins, influencing their structure and function, stabilising protein conformation and preventing protein denaturation within the cell membrane. For oxidative disinfectants, such as hydrogen peroxide and hypochlorite, ROS cause cellular damage. Polyamines have antioxidant properties and may mitigate ROS damage by disinfectants, which contributes to reduced susceptibility and resistance. Finally, polyamines are known to have major impacts on the gene expression of key genes in the bacterial stress response, including *rpoS*, which promotes persister cell formation [45], *oxyR* and *soxRS*, and the subsequent expression of *ahpC*, *katG*, and *katE*, which are integral in resistance to oxidative stress [47,50]. By degrading these molecules, the expression of these genes by polyamines helps maintain cellular integrity and prevent damage to DNA, proteins, and lipids [45].

Links between polyamines and the stress response are evident, as they function as stress modulators [50]. Furthermore, the stress response has been shown to reduce susceptibility to antibiotics [51,52,53]. This work has taken this notion further, confirming the link between the stress response and susceptibility to QAC-disinfectant compounds. This is supported by the changes in susceptibility profiles after stressors and gene expression analysis indicating polyamines as the mediators of susceptibility changes to disinfectants. Although some work has been completed on polyamine effects and antibiotics [32], this has not been the case for disinfectant resistance and represents a novel molecular mechanism for this group of antimicrobials.

The results obtained from transcriptomic and susceptibility data point to a greater understanding of the disinfectant mode of action and mechanisms of resistance. As the threat of antimicrobial resistance grows, these findings could be used to direct future formulations of disinfectants and other antimicrobials. For example, the development of efflux pump inhibitors [54,55,56,57], which could be used in combination with disinfectants and antimicrobials designed to target the stress response [58].

The results and conclusions of this study are limited to the *Serratia* genus and QAC-based disinfectants during aerobic cultivation. In addition, more stressors could be tested at different contact times and severity to reveal the effect on susceptibility. It is unknown whether similar effects will be seen in other bacteria or with different disinfectant compounds, as the literature on this topic is severely limited. However, future work is highly encouraged to further elucidate the effect of various stressors on disinfectant susceptibility.

## 5. Conclusions

In this study, the most differentially expressed genes for QAC-based Virukill^®^- and DDAC-treated cells were identified in QAC-resistant isolate *Serratia* sp. HRI. The differential gene expression profile uncovered strong links to the stress response in the form of polyamine and methionine synthesis and transport in the bacterial response to QAC treatment. Stressors in the form of osmotic, acidic, alkaline, cold, heat, envelope, and oxidative stress were all found to significantly increase the resistance levels of *Serratia* sp. HRI to the QAC-disinfectant Virukill^®^, indicating a strong role of the stress response in susceptibility and resistance to QAC disinfectants.

## Figures and Tables

**Figure 1 microorganisms-12-02240-f001:**
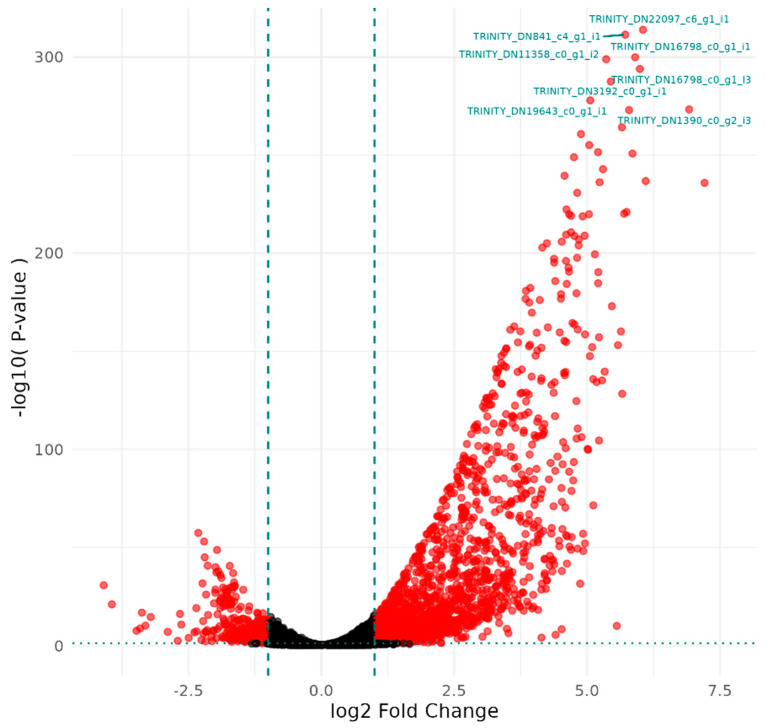
A volcano plot of differentially expressed genes in the DDAC-treated samples plotted by fold change and statistical significance compared to the untreated samples.

**Figure 2 microorganisms-12-02240-f002:**
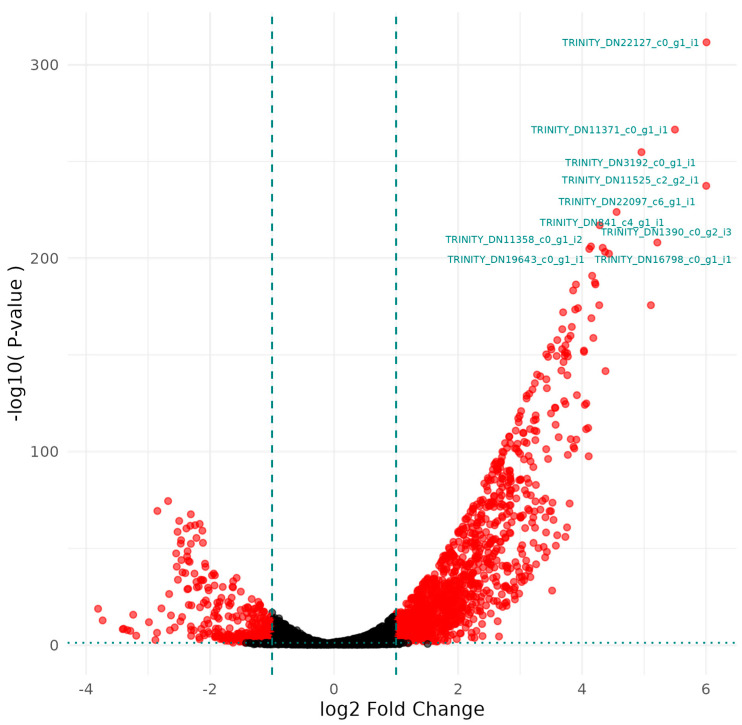
A volcano plot of differentially expressed genes in the Virukill™-treated samples plotted by significance and fold change compared to the untreated samples.

**Figure 3 microorganisms-12-02240-f003:**
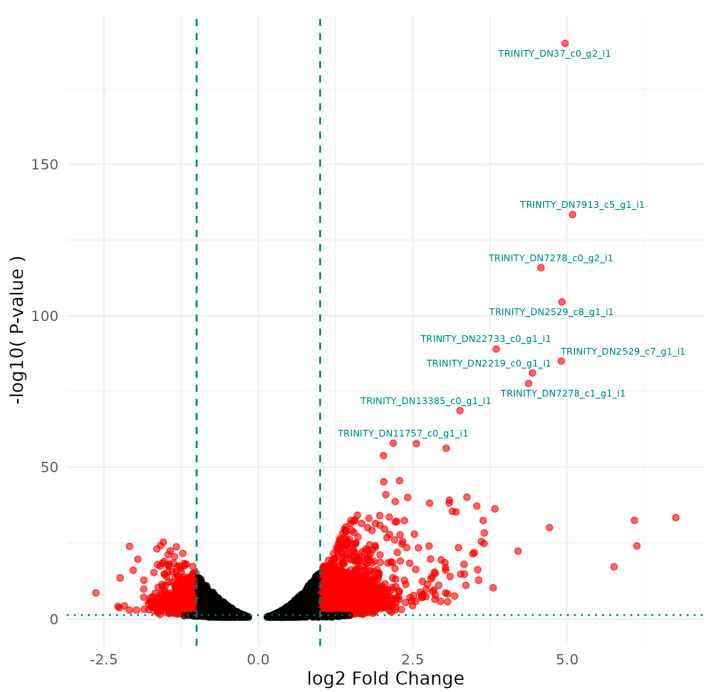
A volcano plot of differentially expressed genes between Virukill™- and DDAC-treated samples plotted by significance and fold change.

**Figure 4 microorganisms-12-02240-f004:**
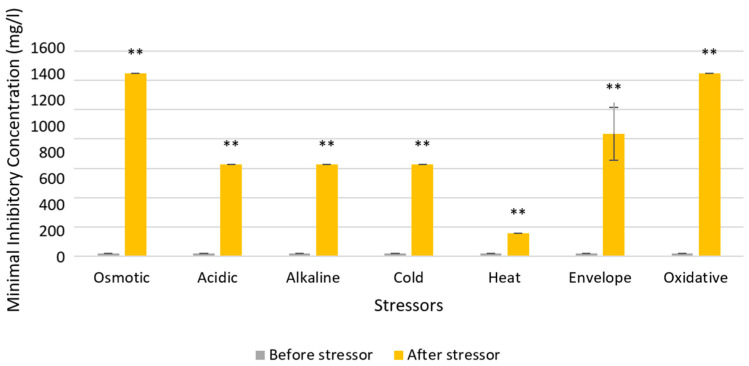
Changes in susceptibility to QAC-based disinfectant Virukill™ for *Serratia* sp. HRI after exposure to one of seven stressors. Statistically significant differences (99% confidence) in MIC after stress condition are labelled with a double asterix.

**Figure 5 microorganisms-12-02240-f005:**
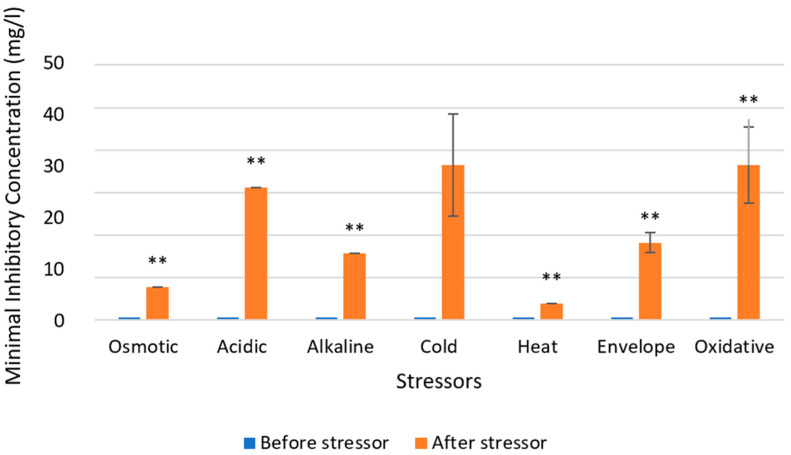
Changes in susceptibility to QAC-based disinfectant Virukill™ for *Serratia marcescens* ATCC after exposure to one of seven stressors. Statistically significant differences (99% confidence) in MIC after stress condition are labelled with a double asterix.

**Table 1 microorganisms-12-02240-t001:** Transcripts with the greatest differential expression (as identified in Figure 1) in DDAC-treated samples compared to untreated control samples annotated by using the UniProt database.

Protein Name	Function
Methionine import ATP-binding protein MetN 2 (EC 7.4.2.11)	A component of the ABC transporter complex MetNIQ involved in methionine import, functions in energy coupling to the transport system. {ECO:0000255|HAMAP-Rule:MF_01719}.
Dihydroxy-acid dehydratase (DAD) (EC 4.2.1.9)	Involved in the biosynthesis of branched-chain amino acids by catalysing the dehydration of (2R,3R)-2,3-dihydroxy-3-methylpentanoate (2,3-dihydroxy-3-methylvalerate) into 2-oxo-3-methylpentanoate (2-oxo-3-methylvalerate) and of (2R)-2,3-dihydroxy-3-methylbutanoate (2,3 dihydroxyisovalerate) into 2-oxo-3-methylbutanoate (2-oxoisovalerate), the penultimate precursor to L-isoleucine and L-valine, respectively. {ECO:0000255|HAMAP-Rule:MF_00012}.
Inducible ornithine decarboxylase (EC 4.1.1.17)	The first enzyme leading to putrescine and thus polyamine synthesis. {ECO:0000305}.
Glycerol dehydrogenase (GDH) (GLDH) (EC 1.1.1.6)	Catalyses the NAD-dependent oxidation of glycerol to dihydroxyacetone (glycerone, allowing for the utilization of glycerol as a carbon source. Additionally, responsible for the oxidation of 1,2-propanediol and 2,3-butanediol and the reduction in dihydroxyacetone. {ECO:0000269|PubMed:7635824}.
Translation initiation factor IF-3	IF-3 binds to the 30S ribosomal subunit and results in a shift of the equilibrium between the 50S and 30S subunits for 70S ribosomes, enhancing the availability of 30S subunits and protein synthesis initiation. {ECO:0000255|HAMAP-Rule:MF_00080}.

**Table 2 microorganisms-12-02240-t002:** Transcripts with the greatest differential expression (labelled in Figure 2) between Virukill™-treated samples and untreated control samples as annotated by the Uniprot database.

Protein Name	Function
Small ribosomal subunit protein bS16 (30S ribosomal protein S16)	
Methionine import ATP-binding protein MetN 2 (EC 7.4.2.11)	A component of the ABC transporter complex MetNIQ involved in methionine import, functions in energy coupling to the transport system. {ECO:0000255|HAMAP-Rule:MF_01719}.
Large ribosomal subunit protein uL15 (50S ribosomal protein L15)	Binds to the 23S rRNA. {ECO:0000255|HAMAP-Rule:MF_01341}.
Protein translocase subunit SecY	The central subunit of the protein translocation channel SecYEG. {ECO:0000255|HAMAP-Rule:MF_01465}.
Large ribosomal subunit protein uL2 (50S ribosomal protein L2)	A critical rRNA binding protein, needed for association of the 30S and 50S subunits to form the 70S ribosome, tRNA binding, and peptide bond formation. It may have peptidyltransferase activity. Forms multiple interactions with the 16S rRNA in the 70S ribosome. {ECO:0000255|HAMAP-Rule:MF_01320}.
Dihydroxy-acid dehydratase (DAD) (EC 4.2.1.9)	Plays a role in the biosynthesis of branched-chain amino acids, catalyses the dehydration of (2R,3R)-2,3-dihydroxy-3-methylpentanoate (2,3-dihydroxy-3-methylvalerate) into 2-oxo-3-methylpentanoate (2-oxo-3-methylvalerate) and of (2R)-2,3 dihydroxy-3-methylbutanoate (2,3-dihydroxyisovalerate) into 2-oxo-3-methylbutanoate (2-oxoisovalerate), the penultimate precursor to L-isoleucine and L-valine, respectively. {ECO:0000255|HAMAP-Rule:MF_00012}.

**Table 3 microorganisms-12-02240-t003:** Transcripts with the greatest differential expression (as labelled in Figure 3) between DDAC-treated and Virukill™-treated samples as annotated by the Uniprot database.

Protein Name	Function
Inducible ornithine decarboxylase (EC 4.1.1.17)	The first enzyme leading to putrescine and thus polyamine synthesis. {ECO:0000305}.
Putrescine transporter PotE (Putrescine-proton symporter/putrescine-ornithine antiporter)	Catalyses both the uptake and excretion of putrescinedependent on the membrane potential and putrescine-ornithine antiporter activity, respectively. {ECO:0000255|HAMAP-Rule:MF_02073}.
Inducible lysine decarboxylase (LDC) (EC 4.1.1.18)	
Dipeptide and tripeptide permease C (Dipeptide/tripeptide: H (+) symporter DtpC)	Proton-dependent permease that transports di- and tripeptides, with significantly higher specificity towards dipeptides. A further preference exists for dipeptides with a C-terminal Lys residue. This protein can bind Ala-Lys, Lys-Ala, Ala-Ala, and transport alanine and trialanine. {ECO:0000269|PubMed:19703419, ECO: 0000269|PubMed: 21933132, ECO: 0000269|PubMed: 22940668, ECO: 0000269|PubMed: 24440353}.
PTS system trehalose-specific EIIBC component (EIIBC-Tre) (EII-Tre) [Includes: Trehalose-specific phosphotransferase enzyme IIB component (EC 2.7.1.201) (PTS system trehalose-specific EIIB component); Trehalose permease IIC component (PTS system trehalose-specific EIIC component)]	The phosphoenolpyruvate-dependent sugar phosphotransferase system (sugar PTS), a major carbohydrate active transport system, catalyses the phosphorylation of incoming sugar substrates concomitantly with their translocation across the cell membrane. This system is involved in trehalose transport at low osmolarity. {ECO:0000269|PubMed:2160944, ECO: 0000305|PubMed:7608078}.
Glycerol dehydrogenase (GDH) (GLDH) (EC 1.1.1.6)	Catalyses the NAD-dependent oxidation of glycerol to dihydroxyacetone (glycerone, allowing for the utilization of glycerol as a carbon source. Additionally, responsible for the oxidation of 1,2-propanediol and 2,3-butanediol and the reduction in dihydroxyacetone. {ECO:0000269|PubMed:7635824}.
Aspartate ammonia-lyase (Aspartase) (EC 4.3.1.1)	
Ascorbate-specific PTS system EIIC component (Ascorbate-specific permease IIC component UlaA)	Functions as a catalyst for the phosphorylation of incoming sugar substrates simultaneously with their translocation across the cell membrane. Specifically, the enzyme II UlaABC PTS system is involved in ascorbate transport. {ECO:0000250|UniProtKB:P39301}.
Nitrate reductase molybdenum cofactor assembly chaperone NarJ (Redox enzyme maturation protein NarJ)	The chaperone required for correct molybdenum cofactor insertion and final assembly of the membrane-bound respiratory nitrate reductase 1. {ECO:0000250}.

## Data Availability

All data will be made available on request.

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
