# Peer review of "Impact of the Stress Response on Quaternary Ammonium Compound Disinfectant Susceptibility in Serratia Species"

_microorganisms, 2024, doi:10.3390/microorganisms12112240_

Round 1

Reviewer 1 Report

Comments and Suggestions for Authors

In this study, the authors examined the impact of Virukill and DDAC on differential gene expression in Serratiaspecies. Additionally, the study confirmed that exposure to stressors can alter the susceptibility of Serratia sp. HRI and Serratia marcescens ATCC isolates to the QAC-based disinfectant Virukill. While the manuscript presents intriguing data, several areas require further clarification in the introduction, materials and methods, results, and discussion sections.

Introduction:

  1. The introduction should be expanded to include more detailed background information, emphasizing the significance of this study.

Materials and Methods:

  1. The authors need to provide a clear rationale for selecting Serratia sp. as the focus of this study.
  2. The study currently includes only two isolates; additional isolates should be examined to strengthen the conclusions.
  3. In line 61, ensure that the organism names are italicized.
  4. In line 65, the text mentions "three disinfectants," but only two are listed. The third disinfectant should be identified, or the text should be corrected if only two were used.
  5. The authors should explain why both Virukill and DDAC were used, given that they are chemically identical.
  6. The methodology for determining the sub-MIC levels of DDAC and Virukill needs clarification, especially regarding the significant difference in sub-MIC levels (76.62 mg/l vs. 1200 mg/l) despite their chemical similarity.
  7. The RNA-Seq methodology should be condensed in the main manuscript, with a detailed workflow provided in the supplementary information.

Results:

  1. The RNA-Seq sequencing data should be submitted to a public repository to enhance transparency and reproducibility.
  2. For Figures 1, 2, and 3, raw expression data for untreated and DDAC/Virukill-treated samples should be provided in an Excel file as supplementary information to aid in result interpretation.
  3. In Tables 1, 2, and 3, the differentially expressed genes vary between DDAC and Virukill. The authors should provide a thorough explanation for these differences, given the chemical similarity of the compounds.
  4. The authors should perform qRT-PCR to validate the differentially expressed genes listed in Tables 1, 2, and 3.
  5. In Figures 4 and 5, changes in susceptibility to DDAC after exposure to seven stressors should also be assessed, considering the different gene activation patterns observed in the RNA-Seq data for Virukill and DDAC.

Discussion:

  1. The discussion should include more detailed information on the differential regulation of Virukill and DDAC, exploring potential reasons for the observed differences in gene expression and their implications.

Author Response

Reviewer 1:

In this study, the authors examined the impact of Virukill and DDAC on differential gene expression in Serratiaspecies. Additionally, the study confirmed that exposure to stressors can alter the susceptibility of Serratia sp. HRI and Serratia marcescens ATCC isolates to the QAC-based disinfectant Virukill. While the manuscript presents intriguing data, several areas require further clarification in the introduction, materials and methods, results, and discussion sections.

Introduction:

The introduction should be expanded to include more detailed background information, emphasizing the significance of this study.

Author response: Corrected- Additional information added.

Materials and Methods:

The authors need to provide a clear rationale for selecting Serratia sp. as the focus of this study.

Author response: Corrected- Added.

The study currently includes only two isolates; additional isolates should be examined to strengthen the conclusions.

Author response: Additional resistant isolates will be acquired for future work.

In line 61, ensure that the organism names are italicized.

Author response: Corrected.

In line 65, the text mentions "three disinfectants," but only two are listed. The third disinfectant should be identified, or the text should be corrected if only two were used.

Author response: Corrected.

The authors should explain why both Virukill and DDAC were used, given that they are chemically identical.

Author response: Corrected-Added.

The methodology for determining the sub-MIC levels of DDAC and Virukill needs clarification, especially regarding the significant difference in sub-MIC levels (76.62 mg/l vs. 1200 mg/l) despite their chemical similarity.

Author response: Corrected- Reference added for the paper where MICs were published together with the methodology. The MIC methodology is as explained in detail in section 2.10.

The RNA-Seq methodology should be condensed in the main manuscript, with a detailed workflow provided in the supplementary information.

Author response: Corrected, methodology condensed in the main manuscript, with a detailed workflow provided in the supplementary information.

Results:

The RNA-Seq sequencing data should be submitted to a public repository to enhance transparency and reproducibility.

Author response: When this project is complete, all raw data and reads will be uploaded to public repositories for free open access and use.

For Figures 1, 2, and 3, raw expression data for untreated and DDAC/Virukill-treated samples should be provided in an Excel file as supplementary information to aid in result interpretation.

Author response: Corrected- Data added in Appendix section.

In Tables 1, 2, and 3, the differentially expressed genes vary between DDAC and Virukill. The authors should provide a thorough explanation for these differences, given the chemical similarity of the compounds.

Author response: Corrected-Additional information added to discussion.

The authors should perform qRT-PCR to validate the differentially expressed genes listed in Tables 1, 2, and 3.

Authors response: Although validation of RNA-Seq has been required in the past, either by qPCR, microarrays, nanobeads etc. Literature and many academics have agreed that the RNA-Seq technology, analysis and assembly pipelines, and databases have advanced to the point where they are robust and accurate enough that qRT-PCR adds little value to RNA-Seq data. For example: Everaert and co-workers (2017) did a systemic analysis and direct comparison between RNA-Seq and qPCR data they found non-concordance in 1.8% of genes. The 1.8% of genes where non-concordance was found had three characteristics in common “risk factors”:

  1. Fold change < 2
  2. Short transcripts
  3. Genes with fewer exon regions

As this paper focused on the most differentially expressed genes and none had the characteristics listed, qPCR would add little to no value to this work.

Some examples of papers that determined good correlations between results obtained with qPCR and with RNA-seq:

  1. Shi Y, He M. Differential gene expression identified by RNA-Seq and qPCR in two sizes of pearl oyster (Pinctada fucata). Gene 2014;538(2):313–22.
  2. Wu AR, et al. Quantitative assessment of single-cell RNA-sequencing methods. Nat Methods 2014;11(1):41–6.
  3. Griffith M, et al. Alternative expression analysis by RNA sequencing. Nat Methods 2010;7(10):843–7.
  4. Asmann YW, et al. 3’ tag digital gene expression profiling of human brain and universal reference RNA using Illumina Genome Analyzer. BMC Genom 2009;10:531.

Additional comments:

Generally, it is not recommended to compare qPCR data with RNA-seq data directly, as they are based on different principles and can be influenced by different biases. qPCR measures the amplification of a specific target sequence, while RNA-seq measures the transcript abundance of all genes in a sample. Additionally, there are differences in sample preparation, library preparation, and sequencing platforms that can affect the data generated by each method. Comparing qPCR CT values and RNA-seq TPM values directly can be problematic because they are based on different scales. qPCR CT values are based on the number of PCR cycles required for the signal to reach a certain threshold, while RNA-seq TPM values represent the relative abundance of a transcript in a sample. Therefore, it is not straightforward to convert between these two values, and it may be difficult to draw meaningful conclusions from such a comparison.

In Figures 4 and 5, changes in susceptibility to DDAC after exposure to seven stressors should also be assessed, considering the different gene activation patterns observed in the RNA-Seq data for Virukill and DDAC.

Author response: Corrected- Additional information added.

Susceptibility profiles and the effect of environmental stressors has been studied on a complete product rather than the active ingredient alone in order to mimic environmental conditions in industry.

Discussion:

The discussion should include more detailed information on the differential regulation of Virukill and DDAC, exploring potential reasons for the observed differences in gene expression and their implications.

Author response: Corrected- Additional information added in discussion.

Reviewer 2 Report

Comments and Suggestions for Authors

In the attachment

Author Response

Response to Reviewer 2:

The manuscript of McCarty and Bragg “Impact of the stresss ...” is interesting, and needs minor changes.

  1. Line 14 abstract should be without acronyms.

Author response: Corrected.

  1. Line 57 add number from ATCC collection.

Author response: Corrected.

  1. Line 62 Type should be with capital letter.

Author response: Corrected.

  1. In the chapters 2.3 and 2.4 add company, country to all media, kit etc.

Author response: Corrected.

  1. Line 70 at first LB acronym should be explained.

Author response: Corrected.

  1. Line 75 at first MIC acronym should be explained.

Author response: Corrected.

  1. Lines 172 and 192 HR and DDCA acronyms should be explained.

Author response: HRI is the isolate strain name, DDAC has been explained in section 2.2.

  1. Figures 1-3 the description of the figures are not very clear.

Author response: Corrected- Results elaborated.

  1. Figures 4 and 5 Serratia should be in italics. Add to Serratia sp.

Author response: Corrected- Figures amended.

Reviewer 3 Report

Comments and Suggestions for Authors

The article comprehensively examines the molecular mechanisms influencing quaternary ammonium compounds disinfectant susceptibility in Serratia species, particularly highlighting the role of polyamines and stress responses. Several areas in the manuscript require improvement.

1.     The title of the manuscript should be written without using abbreviations. It is recommended that "QAC" be replaced with "quaternary ammonium compounds" in the title.

2.     The study's background is insufficiently detailed, and despite selecting Serratia species for their research, the authors have not provided adequate information on Serratia in the introduction.

3.     The research design is not mentioned in the material and methods section.

4.     The rationale for selecting specific stress conditions should be linked to their relevance in practical settings where disinfectants are typically utilized. Furthermore, the controls employed during experiments, particularly in RNA-seq and MIC assessments, could be elaborated upon to strengthen the validity of the findings.

5.     In the discussion section, the implications of these findings for the development of novel antimicrobial agents or disinfectant formulations could be explored in greater depth to underscore the research's real-world applications.

6.     The study limitations have not been addressed, although the study appears to have several.

7.     The conclusion of the manuscript should be free from references and solely based on the study's findings. References may be utilized in the discussion section. Additionally, the conclusion should be revised to focus on the study's specific findings.

8.     The manuscript exhibited a notable deficiency in references from recent studies. It is perplexing that no studies pertinent to the topic appear to have been published in 2024.

Comments on the Quality of English Language

The manuscript should undergo thorough revision to address grammatical and typographical mistakes in several areas of the manuscript.

Author Response

Response to Reviewer 3:

The article comprehensively examines the molecular mechanisms influencing quaternary ammonium compounds disinfectant susceptibility in Serratia species, particularly highlighting the role of polyamines and stress responses. Several areas in the manuscript require improvement.

  1. The title of the manuscript should be written without using abbreviations. It is recommended that "QAC" be replaced with "quaternary ammonium compounds" in the title.

Author response: Corrected.

  1. The study's background is insufficiently detailed, and despite selecting Serratia species for their research, the authors have not provided adequate information on Serratia in the introduction.

Author response: Corrected- Additional information added.

  1. The research design is not mentioned in the material and methods section.

Author response: Corrected- Methodology elaborated.

  1. The rationale for selecting specific stress conditions should be linked to their relevance in practical settings where disinfectants are typically utilized. Furthermore, the controls employed during experiments, particularly in RNA-seq and MIC assessments, could be elaborated upon to strengthen the validity of the findings.

Author response: Corrected- Additional information added.

  1. In the discussion section, the implications of these findings for the development of novel antimicrobial agents or disinfectant formulations could be explored in greater depth to underscore the research's real-world applications.

Author response: Corrected- additional information added.

  1. The study limitations have not been addressed, although the study appears to have several.

Author response: Corrected- limitations added.

  1. The conclusion of the manuscript should be free from references and solely based on the study's findings. References may be utilized in the discussion section. Additionally, the conclusion should be revised to focus on the study's specific findings.

Author response: Corrected- Conclusion rewritten.

  1. The manuscript exhibited a notable deficiency in references from recent studies. It is perplexing that no studies pertinent to the topic appear to have been published in 2024.

Authors response: An additional literature check was done using the PubMed Advanced Search function, no articles were found published in the last two years pertaining to the stress response and QAC susceptibility. As disinfectant resistance is a relatively novel field, literature is limited as most work in the antimicrobial resistance sphere is focused on antibiotics. For example, the methodology for this study was designed after reading a paper published in 2023 doing similar work on antibiotics. Therefore, there are limited references due to the novelty of the work and the growing field.
